# On the 5G and Beyond

**Mário Marques da Silva [1,2,*]** and **João Guerreiro [1,3]**

1   Instituto de Telecomunicações, 1049-001 Lisboa, Portugal; jf.guerreiro@fct.unl.pt
2   Department of Sciences and Technologies, Autonoma University of Lisbon, 1150-293 Lisboa, Portugal
3   Department of Electrical and Computer Engineering, NOVA School of Science and Technology,
    825-149 Caparica, Portugal
*   Correspondence: mmsilva@autonoma.pt; Tel.: +351-213-177-654



**Featured Application: Introductory Article of the MDPI Special Issue "Transmission Techniques for 5G and Beyond".**

**Abstract:** This article provides an overview of the fifth generation of cellular communications (5G) and beyond. It presents the transmission techniques of current 5G communications and those expected of future developments, namely a brief study of non-orthogonal multiple access (NOMA) using the single carrier with frequency domain equalization (SC-FDE) block transmission technique, evidencing its added value in terms of spectral efficiency. An introduction to the sixth generation of cellular communications (6G) is also provided. The insertion of 5G and 6G within the Fourth Industrial Revolution framework (also known as Industry 4.0) is also dealt with. Consisting of a change in paradigm, when compared to previous generations, 5G supports a myriad of new services based on the Internet of things (IoT) and on vehicle-to-vehicle (V2V) communications, supporting technologies such as autonomous driving, smart cities, and remote surgery. The new services provided by 5G are supported by new techniques, such as millimeter waves (mm-wave), in addition to traditional microwave communication, and by massive multiple-input multiple-output (m-MIMO) technology. These techniques were not employed in the fourth generation of cellular communications (4G). While 5G plays an important role in the initial implementation of the Fourth Industrial Revolution, 6G will address a number of new services such as virtual reality (VR), augmented reality (AR), holographic services, the advanced Internet of things (IoT), AI-infused applications, wireless brain–computer interaction (BCI), and mobility at higher speeds. The current research on systems beyond 5G indicates that these applications shall be supported by new MIMO techniques and make use of terahertz (THz) bands.

**Keywords:** 5G; 6G; NOMA; Industry 4.0; massive MIMO; mm-wave; IoT

## 1. Introduction

The Fourth Industrial Revolution considers the replacement of humans by machines in certain tasks, or the development of new or more efficient tasks. Making use of robots and artificial intelligence, the Fourth Industrial Revolution is already deeply modifying society and organizations [1]. As seen in Figure 1 the Fourth Industrial Revolution comprises other parameters besides robots and artificial intelligence [2]. Robots need to communicate and to sense the environment (using sensors and communications), for which the Internet of things (IoT) is employed (all over the Internet protocol (IP)). The IoT generates massive quantities of data (big data) that will be processed with artificial intelligence to generate knowledge; that is, the data supports human decision-making, as well as decisions made by the robots. [3]. These new technologies will originate a deep modification of society with great impact on the human way of life, as well as on the employment market [4].

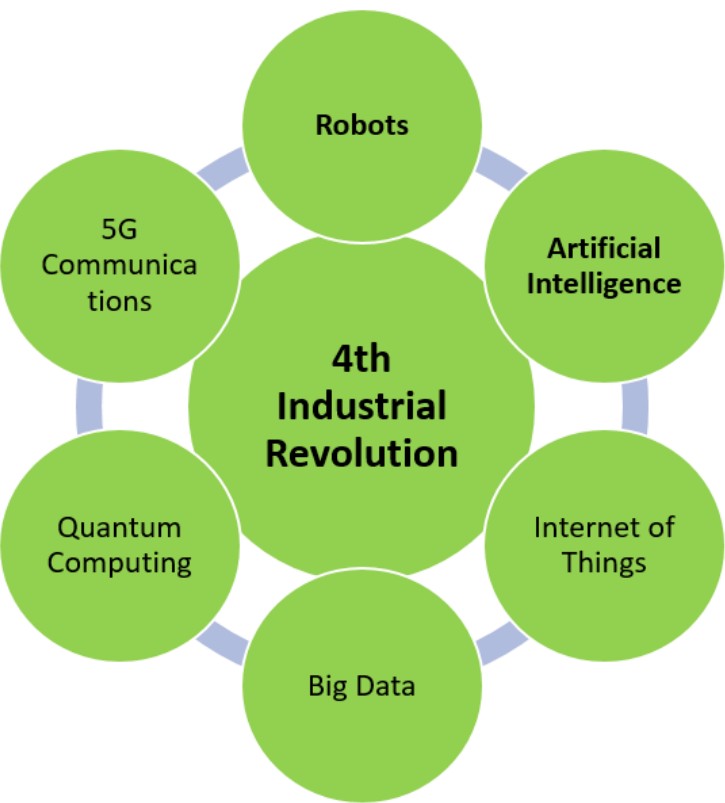

**Figure 1.** The context of the Fourth Industrial Revolution.

Societal modification includes more efficient mobility, based on autonomous cars, smart cities, home safety and automation, intelligent industries, agriculture, smart logistics, medical and lawyer counselling and the use of intelligent drones for a myriad of areas, including in defense (even micro drones used for tactical purposes) [5].

The future of mobility includes autonomous driving [6,7]. A car can be viewed as a robot that uses sensors and communications to interact with the environment, generating a large amount of data and using artificial intelligence to make decisions. Fifth generation (5G) communications play an important key in autonomous driving, using ultra-reliable low-latency communications (URLLC) [8]. As can be observed in Figure 2, URLLC was designed to support new services, such as remote surgery or autonomous vehicles, which are delay-sensitive services that require very low bit error rates (errors being almost unacceptable). 5G communications support vehicle-to-vehicle (V2V) communications without making use of a base station. Point-to-point communications are also considered in other 5G services.

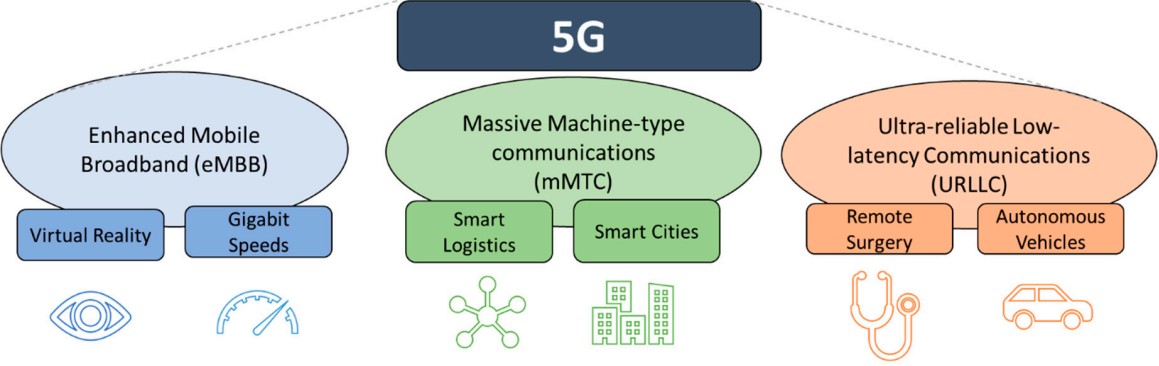

**Figure 2.** Fifth generation (5G) use cases.

Making use of massive multiple-input multiple-output (m-MIMO) technology [9–11] and millimeter wave (mm-wave) communications [12,13] in a large scale, 5G communications provide higher spectral efficiency and capacity than fourth generation (4G) communications [14,15]. It is worth noting that when using carrier frequencies around 60 GHz, instead of the traditional 2 or 3 GHz bands, mm-wave communications facilitate the implementation of m-MIMO technology, as the distance between antennas is greatly reduced and the size of the antennas is also highly reduced. Therefore, both the base station and the mobile terminals can accommodate hundreds of antenna elements [16,17].

Cellular communications and different wireless local area networks (WLANs), namely IEEE 802.11 standards, have used similar technologies [18]. As in 5G communications, the use of m-MIMO and mm-wave technology is also comprised of 802.11ad [12]. Mm-waves are specially well fitted for small cells [19], such as pico cells or femto cells, due to high propagation losses and high coherence bandwidths. In these scenarios, the throughput made available is greatly increased [16,17].

This article aims to provide an overview of 5G and beyond, introducing the transmission techniques for current 5G communications and those expected of future developments. A brief study of non-orthogonal multiple access (NOMA) technology is conducted, viewed as a strong multiple access candidate for new releases of 5G, evidencing its added value in terms of spectral efficiency as it allows multiple users' signals to be transmitted simultaneously in the same carrier frequency. Moreover, sixth generation (6G) communication is also presented. The insertion of 5G and 6G within the Fourth Industrial Revolution is also dealt with. This article is organized as follows: Section 2 describes the fifth generation of cellular communications, while Section 3 presents NOMA, whose performance is studied in multipath fading channels using the single carrier with frequency domain equalization (SC-FDE) block transmission technique. Section 4 describes the perspectives for the sixth generation of cellular communications. Finally, Section 5 concludes this article.

## 2. The Fifth Generation of Cellular Communications

5G new radio (NR) is the new standard of cellular communications [20] and is being built under the performance requirements stated by the International Telecommunication Union (ITU) in 2015 [21]. Contrary to its previous generations, 5G will be much more than just cellular communications and will have different use cases to provide different services. In fact, one of the most interesting features regarding the design of 5G is its flexibility, which is brought about by so-called network slicing [18] and enables different services that go considerably beyond what was offered by its predecessors [22].

As shown in Figure 2, 5G communications involve three main use cases: enhanced mobile broadband (eMBB), massive machine-type communications (mMTC) and URLLC. These use cases will provide very different applications, such as virtual reality, autonomous vehicles, and smart cities, which are key aspects of Industry 4.0. Figure 3 represents the 5G standardization timeline of the Third Generation Partnership Project (3GPP) [23]. The first study item of 5G appeared in 3GPP Release 14, which was still concerned with the development of 4G-Long Term Evolution (4G-LTE).

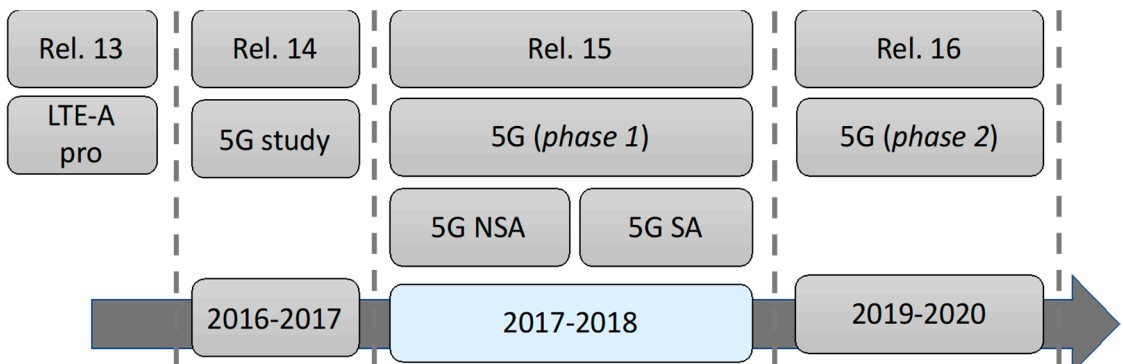

**Figure 3.** Third Generation Partnership Project (3GPP) 5G standardization timeline.

However, the 5G standardization process only started after 3GPP Release 15. In order to accelerate this process, the 3GPP has defined two distinct phases. In the first phase, the focus was the improvement of broadband wireless cellular services (i.e., the eMBB use case) so that they have two distinct modes: the non-standalone (NSA) mode, where the core is still 4G, and the standalone (SA) mode, where the core is 5G. Both modes were standardized during 2018. The second phase, also known as 3GPP Release 16, will be finalized in 2020, and it is concerned with the other 5G use cases, (i.e., mMTC and URLLC use cases).

When it comes to the eMBB use case, substantially higher peak data rates and user-experienced data rates are expected. In fact, the area throughput $R_A$ (measured in bits/s/km$^2$) of 5G will be much higher than in 4G. This substantial increase in $R_A$ will be accomplished by creating a heterogeneous cellular network; in other words, there will be different solutions to increase $R_A$, depending on the cell characteristics. For small cells dedicated to increasing the capacity in a small geographic area, mm-wave communications will be used [24]. Under these circumstances, the area throughput will be augmented by increasing the bandwidth of the communication channels. However, the use of carrier frequencies with tens of gigahertz poses considerable challenges, since the path loss in mm-wave communications is very high. 5G will overcome this problem with m-MIMO techniques, such as beamforming. The use of beamforming leads to large array gains at reception so that the signal-to-noise ratio (SNR) of the received signals is not too degraded. Nevertheless, it should be noted that the use of mm-wave carriers is beneficial in terms of cell density. In fact, as a large path loss yields lower interference levels, cell density can be increased by reducing the inter-base station distance, increasing the system's capacity. For large cells responsible for the coverage and mobility tiers, there is no available bandwidth to increase the capacity. Therefore, massive MIMO techniques will be employed, since they allow for huge gains in spectral efficiency [25]. These gains come from the use of appropriate precoding and decoding schemes that can take advantage of the large number of antennas and yield a very large array and multiplexing gains. This technique is also known as multi-user multiple-input multiple-output (MIMO), and it enables more users to be multiplexed in a given time-frequency resource, which leads to a higher throughput. To summarize, it is evident that the very large capacity gains expected in International Mobile Telecommunications 2020 (IMT-2020) will be attained as a result of the use of different techniques, among which mm-wave, beamforming, and massive MIMO are considered the most important. Note that IMT-2020 corresponds to the future 5G version standardized by the International Telecommunications Union (ITU).

Regarding mMTC communications [26], the ITU has defined strict requirements in terms of the number of connected devices and autonomy. More precisely, the number of connected devices should be one million per square kilometer, and each device should have up to 10 years of autonomy or more. URLLC [8] will enable the development of large sensor networks where the nodes can communicate with very little human interaction. The ITU has defined very ambitious requirements for URLLC in terms of latency. More concretely, the radio interface latency should be just 1 ms, which involves a tenfold reduction compared with the 10 ms achieved in 4G communications. In that context, multiple access will be redesigned in 5G to have higher flexibility [27]. Figure 4 summarizes some of the performance requirements of 5G, showing a comparison with 4G communications.

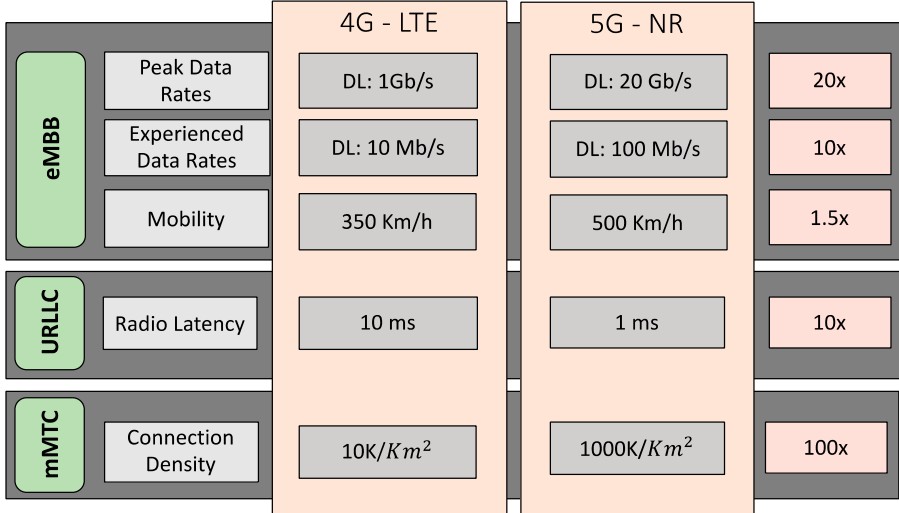

**Figure 4.** Performance requirements of 5G new radio (NR).

As in 4G, the multiple access of 5G communications is based on orthogonal frequency division multiple access (OFDMA) for both the downlink and uplink directions. However, contrary to what happens in 4G, the OFDMA design is very flexible, since different use cases with different requirements should be addressed. As will be described in Section 3, NOMA is viewed as a strong multiple access candidate for the new releases of 5G due to its increased spectral efficiency.

One of the new features of 5G NR is the scalable frequency spacing of OFDMA subcarriers. More precisely, although the subcarrier spacing was $\Delta f = 15$ kHz in 4G, the subcarrier frequency in 5G is given by $\Delta f = 2^\mu \times 15$ kHz, where $\mu$ can vary from $\mu = 0$ to $\mu = 5$ (the maximum subcarrier spacing being $\Delta f = 480$ kHz). Under these conditions, the transmitted waveform can be adapted to the channel conditions. This is known as 5G numerology, and it is very important in 5G since, as the carrier frequency can vary substantially (i.e., both microwave and mm-wave spectrums will be used), the channel conditions can also vary. For instance, when the multipath is strong, a lower subcarrier spacing is desirable. However, when the phase noise is large (which happens frequently in mm-wave communication), a large $\Delta f$ is desirable. A variable $\Delta f$ is also important to support communications where the latency is critical (i.e., for the URLLC use case), since the duration of the OFDMA symbol is inversely proportional to the subcarrier spacing; that is, $T_{OFDM} = \frac{1}{\Delta f}$.

The duration of the physical resource block (PRB), which is the smallest quantity of time-frequency that can be granted to a given user, is 12 subcarriers and one time slot. When $\Delta f = 15$ kHz, there is only one slot per sub-frame (whose duration is always 1 ms). However, when the subcarrier spacing increases, the duration of each OFDMA symbol decreases, which allows for accommodating more slots in a given sub-frame. The PRBs are managed by the base station regularly so that system capacity and performance are balanced, and spectral efficiency is maximized. Table 1 shows some cases of 5G numerology.

**Table 1.** 5G Numerology examples.

| $\Delta_f$ | PRB (Frequency) | PRB (Time) | Slots Per Sub-Frame | Sub-Frame Duration |
|---|---|---|---|---|
| 15 KHz | 180 KHz | 1 ms (eMBB) | 1 | 1 ms |
| 30 KHz | 360 KHz | 0.5 ms | 2 | 1 ms |
| 60 KHz | 720 KHz | 0.25 ms | 4 | 1 ms |
| 120 KHz | 1440 KHz | 0.125 ms (URLLC) | 8 | 1 ms |

Clearly, $T_{OFDM}$ decreases as $\Delta f$ increases, although the sub-frame duration is constant. This means that the number of available PRBs in a given sub-frame can also vary, which explains the very high flexibility of 5G.

Future updates to 5G communications require improved spectral efficiencies. To achieve such a requirement, the inclusion of NOMA is highly foreseeable. NOMA is defined in the following section, together with some results.

## 3. Non-Orthogonal Multiple Access

By exploiting different power levels, NOMA aims to serve multiple users using the same time and frequency [28–33], leading to an improved spectral efficiency when compared with OFDMA [29–32]. Since MIMO systems are a key feature of 5G, the inclusion of NOMA always has to be incorporated with MIMO [33], or more precisely with m-MIMO.

NOMA is an effective mechanism to accommodate a higher number of users without a spectrum increase. This leads to a higher channel capacity, especially useful in scenarios with extremely high numbers of mobile terminals, such as in 5G use cases with mMTC or URLLC [29,31,32].

Due to the near–far problem and power control, the transmission power of different users suffers variations. By employing successive interference cancellation (SIC) and detecting users' signals in descending order, NOMA can detect different users that share the same time and frequency. Two different types of NOMA exist: conventional NOMA and cooperative NOMA. As can be viewed in Figure 5 with conventional NOMA, the SIC of a reference user only detects, regenerates, and cancels users' signals with powers higher than the reference user. Therefore, those user signals closer to the base station (with less power, due to power control) are not cancelled, representing interference, and degrading the performance (see User 1 in Figure 5). On the other hand, cooperative NOMA allows the cancellation of all interfering users' signals and provides diversity. Cooperative NOMA considers that users closer to the base station, that have previously detected and subtracted more powerful users (farther from the base station), send over the air copies of signals of said more powerful users. Let us focus on the example depicted in Figure 5. User 2, the user closer to the base station (strong user, due to stronger channel conditions), using SIC, detects User 1 first (a weak user, i.e., a more powerful user) and subtracts this signal from the received signal before detecting its own signal (User 2's signal). Assuming cooperative NOMA, User 2 sends over the air the previously detected signal of User 1. While with conventional NOMA User 1's signal detection would be corrupted by User 2's signal (less powerful and therefore not detected by SIC), with cooperative NOMA, User 1 also receives the copy of its signal sent by User 2, and then employs an efficient algorithm to combine the two versions of User 1's signal. Note that the version of User 1's signal detected by User 2 should be subject to the cancellation of User 2's signal before it is transmitted. This is an action that allows such a copy of User 1's signal to be clean of interference. Therefore, cooperative NOMA brings special added value for users that are farther from the base station) i.e., those with higher powers).

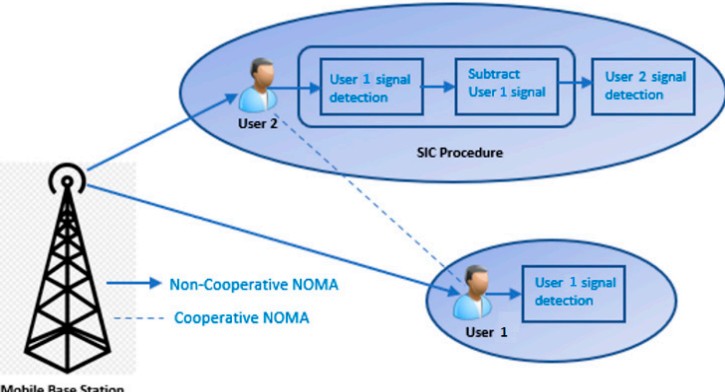

**Figure 5.** Illustration of a non-orthogonal multiple access (NOMA) scheme, considering a two-user scenario.

With the aim of explaining the NOMA concept, rather than performing a detailed scientific study, Figure 6 shows the performance results for conventional NOMA (designated in the figure as NOMA) and cooperative NOMA (designated in the figure as COOP NOMA) with $4 \times 32$ MIMO, considering the zero forcing receiver, in the downlink direction. Similar to Figure 5, two NOMA users were considered in the simulation with received powers of (1, 0.5), where the first value (1 in this case) corresponded to the power of the reference user and the other value was the power of the interfering user (0.5 in this case, i.e., 3 dB below). Considering the downlink transmission, the base station sent the two user signals superimposed, but with different powers. With NOMA, users with poorer channel conditions are allocated more power. Therefore, due to the near–far problem, a user closer to the base station tends to receive signals with less power (0.5 in this case) than those farther from it (1 in this case). Moreover, users located at the edge of the cell are more subject to inter-cell interference, requiring more power to keep the signal-to-noise ratio at an acceptable level. Naturally, the fading effects, and the power control used to mitigate these effects, can modify these power scales. We can associate the two users considered in Figure 6 to the scenario depicted in Figure 5. The reference user, with a power of 1, corresponds to User 1 of Figure 5 (farther from the base station), and the interfering user, with a power of 0.5, corresponds to User 2 of Figure 5 (closer to the base station). Let us focus on the scenario employed in Figure 6. Considering that the propagation losses occurring in the distance between the location of User 2 and the location of User 1 is PATH_LOSS, at User 2's location (the reference user) the received power of User 1 is 1, and the received power of User 2 is 0.5. Similarly, at User 1's location (the interfering user), the power of 1 is 1 + PATH_LOSS (PATH_LOSS having a positive effect), while the power of User 2 is 0.5 + PATH_LOSS (PATH_LOSS having a positive effect). Consequently, regardless of the considered user (reference user or interfering user), the difference of powers between them is kept unchanged.

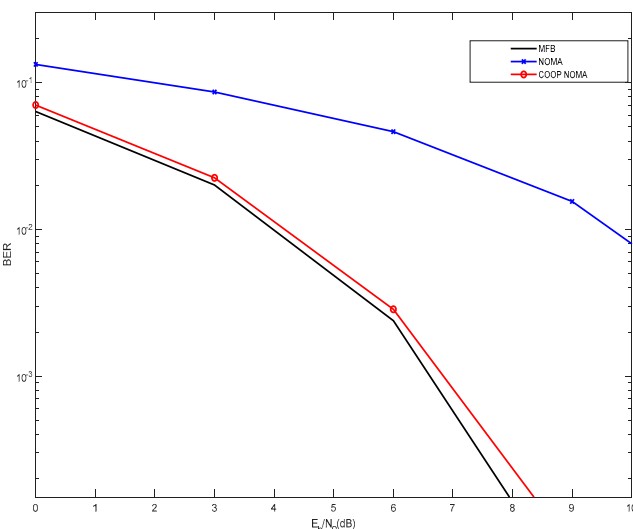

**Figure 6.** Results for two NOMA users with powers of (1, 0.5), with $4 \times 32$ MIMO.

The performance was evaluated using Monte Carlo simulations, observing the bit error rate (BER) as a function of $E_b/N_0$, where $E_b$ is the energy of the transmitted bits and $N_0$ is the one-sided power spectral density of the noise. Ideal channel estimation and the SC-FDE were assumed [11,15], with quaternary phase shift keying (QPSK) modulation and a block length of N = 256 symbols (with similar results observed for other values of N, provided that N > 1). A Rayleigh fading channel was considered, with 16 uncorrelated equal power paths. The duration of the useful part of the blocks (N symbols) was 1 μs, and the cyclic prefix had a duration of 0.125 μs. The match filter bound (MFB) curve is a way to measure the channel modeled by the sum of the delayed and independently Rayleigh-fading rays, which can be viewed as a lower bound [34].

As seen in Figure 6, the results with conventional NOMA are quite limited, due to the existence of a high level of interference. Note that the SIC that is part of the receiver of the reference user (User 1 of Figure 5) only detects, regenerates, and cancels user signals with powers higher than that of the reference user, which is not the case here as the interfering user has a power of 0.5, which is not cancelled. This explains the low performance achieved with conventional NOMA.

Cooperative NOMA comprises the detection of other users with SIC on the receiver of the interfering users (closer to the base station and, therefore, with lower power). Therefore, the interfering signals associated with all users are cancelled. The power of User 2 (seen by User 1 as an interfering user) is higher than that of User 1, being detected, regenerated, and cancelled using SIC to allow the detection of the data sent to User 2. Cooperative NOMA considers that User 2 retransmits the symbols detected by SIC (typically using time-division multiplexing). These two signals are combined to improve performance. In Figure 6, we can see that the combination of signals performed with cooperative NOMA results in a high level of performance improvement, as compared with conventional NOMA. In fact, the performance of cooperative NOMA is very close to the MFB.

Figure 7 shows the BER performance in the same scenario as that of Figure 6, the only difference being that the power of users is (0.5, 1) (instead of (1, 0.5)). In this scenario, the reference user is closer to the base station (power of 0.5), while the interfering user is farther from the base station (power of 1). What is noteworthy is that the NOMA receiver comprises the detection, regeneration, and cancellation of the users' signals by a descending order of powers (up to the power of the reference user) before the reference user is detected. Consequently, the detection of the reference receiver is clean of interferences, and the performance achieved with conventional NOMA is already good. In this scenario, since the reference user is the one closer to the base station and with a power lower than all others, such signal detection is not carried out in the SIC receiver of other users, and cooperative NOMA is not implemented as its performance would be the same as conventional NOMA.

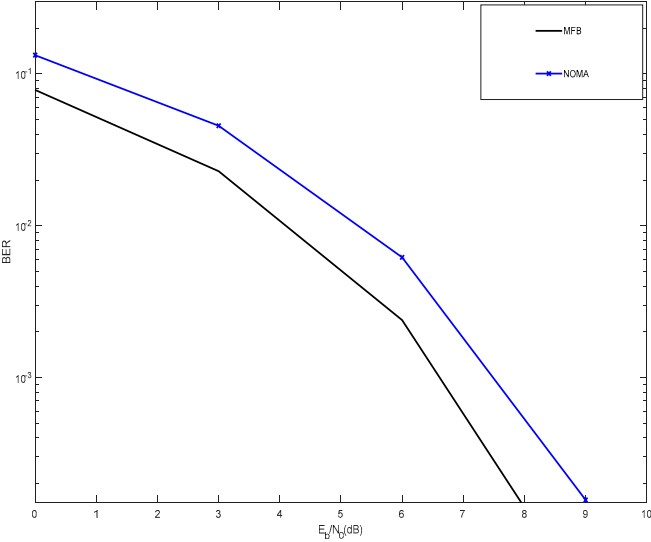

**Figure 7.** Results for two NOMA users with powers of (0.5, 1), with $4 \times 32$ MIMO.

Future research on NOMA will include performance evaluation in different user scenarios (more users, power levels, etc.).

## 4. The Sixth Generation of Cellular Communications

In terms of communications, 5G comprises a modification of a paradigm, focusing on the initial requirements of the Fourth Industrial Revolution and its implications in organizations and society. Nevertheless, new services and requirements continue to develop and, consequently, the demands from communications are always increasing [35–37]. The future digital society, in the scope of increasing

automation, namely the digital society of 2030 and beyond, comprises more and more connected devices (IoT), including sensors, vehicles, aerial drones, and data. While 5G supports autonomous vehicles, the increasing number of sensors per vehicle requires higher speeds of communications and lower latencies. Society and organizations demand new services to be included in 6G, such as (see Figure 8) the following:

- Augmented reality (AR) and extended reality (XR);
- Artificial intelligence (AI)-infused applications;
- Wireless brain–computer interactions (BCI);
- Holographic services;
- The integration of communications with localization, mapping, and remote control;
- Emerging eHealth applications;
- Improved autonomous vehicles;
- More efficient support of IoT, namely smart cities and smart houses, supporting an extremely high number of low-power devices;
- Support of flying vehicles and increased mobility speed.

In addition, 6G aims to have higher energy efficiency and more efficient strategies of energy-harvesting, so that the autonomy of user equipment can be increased, despite its demanding applications.

**Figure 8.** Evolution of cellular generations.

These new services and capabilities to be supported by 6G continue to require more efficient networks with increased data rates, lower latencies, more efficient spectral efficiencies, increased energy efficiencies, and improved network capacities. Some of the foreseen requirements for 6G include:

- Nomadic peak data rates of at least 1 Tbps (100 times higher than 5G);
- Mobile data rates of 1 Gbps (10 times higher than 5G);
- An energy efficiency 10 to 100 times better than 5G;
- A spectral efficiency 5 to 10 times better than 5G.

While 5G requirements are achieved based on mm-wave and m-MIMO techniques, 6G must incorporate new concepts and frequency bands not yet considered for cellular communications.

This includes visible light communications (VLC) and terahertz bands (100 GHz–10 THz) [38], enabling data rates in the order of hundreds of Gbps. VLC is a mature communication technique well suited for short-range coverage, though it is susceptible to interference, such as that from the sun. On the other hand, flying vehicles, such as drones, alongside a communications paradigm based on heterogenous networks (conventional cells, vehicle-to-everything communications (V2X), IoT, drones, balloons, satellites, etc.), will require a three-dimensional (3D) network architecture with 3D coverage, instead of two-dimensional, as considered by 5G. Mobility speeds of up to 1000 km/h are also expected to be a requirement of 6G.

## 5. Conclusions

In the scope of the Fourth Industrial Revolution, humans are being replaced by robots in several tasks. It aims to achieve more efficient mobility based on, among other things, self-driving cars, smart cities, intelligent industries, medical and lawyer counselling, and the use of intelligent drones for a plethora of areas, including in defense.

Making use of mm-wave communications, massive MIMO, beamforming, and device-to-device communications, 5G communications are an enabler of the Fourth Industrial Revolution. Among the wide range of services supported by 5G communications, one can refer smart agriculture, smart cities, IoT, eHealth (using mMTC), autonomous vehicles, factory automation, remote surgeries (using URLLC), or providing wide coverage with an improved throughput several times higher than with 4G (using eMBB). Support a higher speed of communications and lower latency than 4G, as well as point-to-point communications, 5G is an important facilitator of the Fourth Industrial Revolution.

NOMA is a strong multiple access candidate for new 5G releases, especially well-suited for mMTC. It was briefly shown that NOMA, using the block transmission technique SC-FDE and in the presence of a strong multipath fading channel, brings added value in terms of spectral efficiency as it allows multiple users' signals to be transmitted simultaneously in the same carrier frequency. Future research on NOMA will include performance evaluation in different user scenarios (more users, power levels, etc.).

New services and requirements continue to advance and, consequently, the demands for communications increase. The future digital society, in the scope of increasing automation, namely the digital society of 2030 and beyond, comprises more and more IoT devices, including sensors, vehicles, aerial drones, and data. An advanced enabler of the Fourth Industrial Revolution, 6G will support higher communication speeds and lower latencies as well as new services, such as AR, XR, AI-infused applications, advanced IoT, BCI, holographic services, and mobility at higher speeds.

**Author Contributions:** Both authors have contributed equally to the article. Both authors have read and agreed to the published version of the manuscript.

**Funding:** This work was funded by FCT/MCTES through national funds and, where applicable, co-funded EU funds under project UIDB/50008/2020.

**Conflicts of Interest:** The authors declare no conflict of interest.

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
