# Peer review of "On the 5G and Beyond"

_applsci, doi:10.3390/app10207091_

Round 1
Reviewer 1 Report
It seems that this paper intends to provide some sort of overview of 5G communications, with special focus on NOMA and the intended use cases for this new technology. However, in my opinion, the review is not sufficiently comprehensive to be useful to the readers and the NOMA simulations are too simple to justify any meaningful conclusions.
First of all, the intent of the paper is never clear. Neither the title, abstract or introduction specify what the paper is going to do, they seem like a soup of buzzwords without a connecting argument.
If this is a review article, it should have a lot more and better references. Many topics that the authors describe as very important are introduced without a reference, e.g. network slicing, pico and femto-cells.
The writing also needs significant improvement. There are many syntax, grammar and punctuation errors. I will include a few of them as minor comments below, but there are many others.
I find that the introduction is poorly structured. It keeps jumping from one topic to another without goal. Eg, second paragraph starts with drones, then cars, then robots, then cars again, then communications. Furthermore, NOMA is not mentioned in the abstract or Intro, despite it constitutes about half of the body of the paper.
Many claims need to be justified with references. For example P4L125, the strict requirements that the ITU has defined.
Cooperative NOMA works well for the two user toy example studied in the paper. What if there are many users? The paper should at least discuss that case.
The powers used to generate figure 6 are (1, 0.5). Are those received or transmit powers? What distance is each user from the base station? Are you considering the fact that each user will hear each signal with different power? In any case, a 3dB difference in power is not very realistic to illustrate the near-far problem in NOMA. Also, how strong is the fading? Specifying the distribution (Rayleigh) and the number of paths is not enough.
Give a reference or explain how the MFB bound works.
Minor comments:
1.- Abstract: "Holographic services" appears twice in the enumeration of 6G applications
2.- Last sentence in page: as well AS
3.- Intro: "which translates INTO"
4.- P2L64: no comma after "it is worth noting that"
5.- P3L73: Nevertheless means "in spite of", it needs a contradiction.
5.- The intro talks about throughputs, in plural. It is unclear why there are multiple throughputs.
6.- P4L121 Define and give reference for IMT-2020
7.- P4L127 It is unclear which use case the paper is referring to
8.- P4L128 URLLC is a whole section of 5G, not a use case
9.- P5L157 so THAT the system
10.- P6L169 translates TO an improved
11.- P6L181 explain what you mean by reference
12.- Many times in section 3, the term farer is used. That is wrong, it should be farther
13.- P6L196 user 1 signal TO BE clean of
14.- The caption in the figures are hard to read, too small font.
15.- P10L288: Talking about aerial cars to justify the need for 6G is beyond unrealistic. I don´t think a scientific article, specially one focusing on the applications of an upcoming technology should be mixed with science fiction.
Author Response
Dear Sir,
This is a revised version of the Manuscript applsci-932293 entitled ”On the 5G and Beyond”, by M. Marques da Silva, and João Guerreiro.
We would like to thank the reviewer for the helpful comments. We believe we addressed all issues raised by the reviewer in the revised version of the manuscript. We include a detailed answer to the reviewer’s comments.
Kindest regards,
Mario Marques da Silva
João Guerreiro
Referees' Comments to Author with Responses
It seems that this paper intends to provide some sort of overview of 5G communications, with special focus on NOMA and the intended use cases for this new technology. However, in my opinion, the review is not sufficiently comprehensive to be useful to the readers and the NOMA simulations are too simple to justify any meaningful conclusions.
First of all, the intent of the paper is never clear. Neither the title, abstract or introduction specify what the paper is going to do, they seem like a soup of buzzwords without a connecting argument.
Authors’ Reply: We agree that the abstract and introduction could have made clearer. These sections were modified to clarify the main purpose of the paper.
If this is a review article, it should have a lot more and better references. Many topics that the authors describe as very important are introduced without a reference, e.g. network slicing, pico and femto-cells.
Authors’ Reply: This comment was taken into account in the revised document and references were added/modified.
The writing also needs significant improvement. There are many syntax, grammar and punctuation errors. I will include a few of them as minor comments below, but there are many others.
Authors’ Reply: Thank you for pointing out the errors. A review of the wording was carried out.
I find that the introduction is poorly structured. It keeps jumping from one topic to another without goal. Eg, second paragraph starts with drones, then cars, then robots, then cars again, then communications. Furthermore, NOMA is not mentioned in the abstract or Intro, despite it constitutes about half of the body of the paper.
Authors’ Reply: This comment was taken into account in the revised document, and the second paragraph was modified accordingly.
Many claims need to be justified with references. For example P4L125, the strict requirements that the ITU has defined.
Authors’ Reply: it was clarified that IMT-2020 corresponds to the future 5G version standardized by the International Telecommunications Union.
Cooperative NOMA works well for the two user toy example studied in the paper. What if there are many users? The paper should at least discuss that case.
Authors’ Reply: As described in the abstract, the purpose of this paper is to act as review paper about 5G and beyond, namely it is an Introductory Article of the MDPI Special Issue “Transmission Techniques for 5G and Beyond”. Therefore, the two user example was only to explain the functionalities of NOMA, and its advantages, as a strong multiple access technique for a future release of 5G, not to make a detailed study of it. This was clarified in the paper. A research paper on the topic is being finalized and will be released soon with all the details and in multiple scenarios.
The powers used to generate figure 6 are (1, 0.5). Are those received or transmit powers? What distance is each user from the base station? Are you considering the fact that each user will hear each signal with different power? In any case, a 3dB difference in power is not very realistic to illustrate the near-far problem in NOMA. Also, how strong is the fading? Specifying the distribution (Rayleigh) and the number of paths is not enough.
Authors’ Reply: It was clarified in the paper that the mentioned powers are received powers. As previous described, being this a review article, the examples are only to show the concept of NOMA, and advantages, rather than performing a detailed study, which will be released in a research article about the subject. The 3 dB difference was chosen to show that NOMA tends to perform well even with such low difference of received powers. Naturally that the fading can create problems with the implementation of the SIC, as part of the NOMA receiver, more precisely cooperative NOMA. Nevertheless, since the two-user scenario is not realistic (normally, more than two users exist), those interfering users whose fading does not alter the received power order can be used as valid users contributing to cooperative NOMA. As described in the paper, a Rayleigh fading channel was considered with 16 uncorrelated equal power paths (it was assumed invariant during the block duration). The duration of the useful part of the blocks (N symbols) is 1μs and the cyclic prefix has duration 0.125μs. The impact of the CP duration in the performance is residual as long as the impulse response presents a high number of separated multipath components (with different delays with regard to the symbol period), which is the case of the current paper.
Give a reference or explain how the MFB bound works.
Authors’ Reply: This comments was taken into account in the revised document.
Minor comments:
Authors’ Reply: These minor comments were taken into account in the revised manuscript.
Reviewer 2 Report
This review manuscript introduces 5G and its associated content. It clarifies the main advantages that distinguish 5G from 4G, and provides some perspective on 6G. This article serves as an introduction to 5G.
Minor comment: It is better to provide a brief about the "fourth industrial revolution"
Author Response
Dear Sir,
This is a revised version of the Manuscript applsci-932293 entitled ”On the 5G and Beyond”, by M. Marques da Silva, and João Guerreiro.
We would like to thank the reviewer for contributing to this paper, with the review. We believe we addressed all issues raised by the reviewer in the revised version of the manuscript.
Kindest regards,
Mario Marques da Silva
João Guerreiro
Reviewer 3 Report
The paper's composition is coherent; the structure is logical and meets the goal of the paper. The title " On the 5G and Beyond " put well the paper's objective; it is clear and expresses the issue being assessed very well. Conclusions are related to the results presented before reflecting the assessed issue at a professional level. All the figures and tables are complete and understandable. Author uses enough tables and figures featuring a great deal of data being processed hence adding a higher added value to the paper. I found the paper well-written and cohesive. Authors appear to be professionals, very well oriented and involved in the observed issue. The length of the paper is adequate to the significance of the topic. However major revision is necessary to get the manuscript published in the journal. It is recommended the authors to make a revision, and the specific amendments to the text are as follows:
- The goal explicitly stated within the Introduction and Abstract clearly expressing the main problem and purpose of the paper and author's intention being assessed and discussed within the paper along with its clear and unambiguous formulation would be recommended.
- The research results and paper´s outcome should be mentioned in the Abstract.
- In abstract and introduction to underline the added value, purpose and ways of application of the research results is recommended.
- In Introduction part I recommend to mention the way how the research results could be implemented in the practice bringing up any benefits and added value by expressing the research novelty.
- I recommend adding to the conclusion section authors’ further research directions within this explored issue along with a brief research limitation.
Author Response
Dear Sir,
This is a revised version of the Manuscript applsci-932293 entitled ”On the 5G and Beyond”, by M. Marques da Silva, and João Guerreiro.
We would like to thank the reviewer for the helpful comments. We believe we addressed all issues raised by the reviewer in the revised version of the manuscript. We include a detailed answer to the reviewer’s comments.
Kindest regards,
Mario Marques da Silva
João Guerreiro
Referees' Comments to Author with Responses
The paper's composition is coherent; the structure is logical and meets the goal of the paper. The title " On the 5G and Beyond " put well the paper's objective; it is clear and expresses the issue being assessed very well. Conclusions are related to the results presented before reflecting the assessed issue at a professional level. All the figures and tables are complete and understandable. Author uses enough tables and figures featuring a great deal of data being processed hence adding a higher added value to the paper. I found the paper well-written and cohesive. Authors appear to be professionals, very well oriented and involved in the observed issue. The length of the paper is adequate to the significance of the topic. However major revision is necessary to get the manuscript published in the journal. It is recommended the authors to make a revision, and the specific amendments to the text are as follows:
The goal explicitly stated within the Introduction and Abstract clearly expressing the main problem and purpose of the paper and author's intention being assessed and discussed within the paper along with its clear and unambiguous formulation would be recommended.
Authors’ Reply: We agree that the abstract and introduction could have made clearer. These sections were modified accordingly.
The research results and paper´s outcome should be mentioned in the Abstract.
Authors’ Reply: The abstract as modified as recommended
In abstract and introduction to underline the added value, purpose and ways of application of the research results is recommended.
Authors’ Reply: The abstract was modified to take this comment into account.
In Introduction part I recommend to mention the way how the research results could be implemented in the practice bringing up any benefits and added value by expressing the research novelty.
Authors’ Reply: The introduction was modified accordingly.
I recommend adding to the conclusion section authors’ further research directions within this explored issue along with a brief research limitation.
Authors’ Reply: The conclusion section was modified as recommended.
Round 2
Reviewer 1 Report
I still feel that the contribution of this paper to the research community is very minor. It does not serve as a comprehensive review paper neither does it propose any original ideas. However, it could be suitable as an "introductory paper" to a special issue, which seems to be the intended goal. I am not familiar with the expectations of such "introductory papers".
The authors have done a good job addressing most of my prior comments, but there are still a few issues left that I believe should be addressed before publication:
1.- English language and style should be improved, specially if this paper is to be featured prominently in a special issue. I recommend using a professional English editor for this.
2.- Add reference to URLLC standard the first time it is introduced (third paragraph of intro)
3.- 5th paragraph of intro: since throughput was changed to singular, verb should be "is" instead of "are"
4.- I still don't find the experiment illustrated in Figure 6 clear enough. We have two signals being transmitted to two users at different distances from the BS. Presumably, the two signals are transmitted with different powers. What does it mean then that the received powers are 1 and 0.5? Who receives which signal with what power? Are the two signals transmitted with the same power? How is the channel attenuation to each user modeled? Are they the same despite them being at different distances?
5.- Figure legends and axis are hard to read. I suggest increasing font size.
Author Response
Dear Sir,
This is a revised version of the Manuscript applsci-932293 entitled ”On the 5G and Beyond”, by M. Marques da Silva, and João Guerreiro.
We would like to thank the reviewer for the helpful comments. We believe we addressed all issues raised by the reviewer in the revised version of the manuscript. We include a detailed answer to the reviewer’s comments.
Kindest regards,
Mario Marques da Silva
João Guerreiro
Referees' Comments to Author with Responses
I still feel that the contribution of this paper to the research community is very minor. It does not serve as a comprehensive review paper neither does it propose any original ideas. However, it could be suitable as an "introductory paper" to a special issue, which seems to be the intended goal. I am not familiar with the expectations of such "introductory papers". The authors have done a good job addressing most of my prior comments, but there are still a few issues left that I believe should be addressed before publication:
1.- English language and style should be improved, specially if this paper is to be featured prominently in a special issue. I recommend using a professional English editor for this.
Authors’ Reply: English language and style have been revised, as recommended by the reviewer.
2.- Add reference to URLLC standard the first time it is introduced (third paragraph of intro)
Authors’ Reply: Reference [13] has been added, as suggested.
3.- 5th paragraph of intro: since throughput was changed to singular, verb should be "is" instead of "are"
Authors’ Reply: We thank the reviewer for detecting this typo. This was corrected.
4.- I still don't find the experiment illustrated in Figure 6 clear enough. We have two signals being transmitted to two users at different distances from the BS. Presumably, the two signals are transmitted with different powers. What does it mean then that the received powers are 1 and 0.5? Who receives which signal with what power? Are the two signals transmitted with the same power? How is the channel attenuation to each user modeled? Are they the same despite them being at different distances?
Authors’ Reply: The following wording has been added to the paper to make it clearer: “Considering the downlink transmission, the base station sends the two user signals superimposed, but with different powers. With NOMA, users with poorer channel conditions are allocated more power. Therefore, due to near-far problem, a user closer to the base station receives signals with less power (0.5, in this case), than those farther (1, in this case). Moreover, users located at the edge of the cell are more subject to inter-cell interference, requiring more power to keep the signal-to-noise ratio at an acceptable level. Naturally that the fading effects can modify these power scales.”. The basic idea of NOMA is that users with poorer channel conditions receive signals with more power. The power control should be implemented to support such requirement, despite the near-far problem.
5.- Figure legends and axis are hard to read. I suggest increasing font size.
Authors’ Reply: The font size of figure legends have been increased, as recommended.
Reviewer 3 Report
All the necessary observations and comments have been incorporated into the revised manuscript that’s why I recommend this paper to be published in Applied Sciences journal. The revised paper titled “On the 5G and Beyond“ intended to be published in JCMS: Journal of Common Market Studies meets all the requirements for professional scientific journal. All the significant comments, recommendations and remarks of reviewers have been incorporated into the manuscript in a proper way giving the paper higher added value and professional features.
Author Response
Dear Sir,
This is a revised version of the Manuscript applsci-932293 entitled ”On the 5G and Beyond”, by M. Marques da Silva, and João Guerreiro.
We would like to thank the reviewer for the helpful comments. We believe we addressed all issues raised by the reviewer in the revised version of the manuscript. We include a detailed answer to the reviewer’s comments.
Kindest regards,
Mario Marques da Silva
João Guerreiro
Round 3
Reviewer 1 Report
The added language to clarify the experiment illustrated in Figure 5 is still not enough for my understanding.
Lets keep it simple. In the experiment, we have two users U1 and U2, and a transmitter broadcasting two signals S1 and S2. In the simulations that were used to generate Fig. 6,
1.- What was the received power of S1 at U1?
2.- What was the received power of S1 at U2?
3.- What was the received power of S2 at U1?
4.- What was the received power of S2 at U2?
These values need to be specified in the paper to make the experiment repeatable and to make sure that the setup makes sense.
Author Response
Dear Sir,
This is a revised version of the Manuscript applsci-932293 entitled ”On the 5G and Beyond”, by M. Marques da Silva, and João Guerreiro.
We would like to thank the reviewer for the helpful comments. We believe we addressed all issues raised by the reviewer in the revised version of the manuscript. We include a detailed answer to the reviewer’s comments.
Kindest regards,
Mario Marques da Silva
João Guerreiro
Authors’ Reply: In the simulations that were used to generate Fig. 6, we assumed the following power:
- The received power of S1 at U1 is 1
- The received power of S1 at U2 is 1 + propagation losses that do not occur
- The received power of S2 at U1 is 0.5
- The received power of S2 at U2 is 0.5 + propagation losses that do not occur
The following text has been added to the paper: “Let us focus on the scenario employed in Figure 6. Considering that the propagation losses that occur between the difference of distance between location of user 2 and location of user 1 is PATH_LOSS, at user 2 location (the reference user), the received power of user 1 is 1, and the received power of user 2 is 0.5. Similarly, at user 1 location (interfering user), the power of 1 is 1+PATH_LOSS (PATH_LOSS has a positive effect), while the power of user 2 is 0.5+PATH_LOSS (PATH_LOSS has a positive effect). Consequently, regardless of the considered user (reference user or interfering user), the difference of powers between them is kept unchanged.”.